# Slow-Paced Breathing Intervention in Healthcare Workers Affected by Long COVID: Effects on Systemic and Dysfunctional Breathing Symptoms, Manual Dexterity and HRV

**DOI:** 10.3390/biomedicines12102254

**Published:** 2024-10-03

**Authors:** Marcella Mauro, Elisa Zulian, Nicoletta Bestiaco, Maurizio Polano, Francesca Larese Filon

**Affiliations:** 1Unit of Occupational Medicine, Department of Medical Sciences, University of Trieste, 34129 Trieste, Italylarese@units.it (F.L.F.); 2Experimental and Clinical Pharmacology Unit, Centro di Riferimento Oncologico di Aviano, Istituto di Ricovero e Cura a Carattere Scientifico, 33081 Aviano, Italy

**Keywords:** long COVID, breathing exercises, autonomic effects, vagus nerve stimulation, clinical trial, Nijmgen questionnaire, medical unexplained symptoms, executive functions, Purdue Pegboard test, heart rate variability

## Abstract

Background: Many COVID-19 survivors still experience long-term effects of an acute infection, most often characterised by neurological, cognitive and psychiatric sequelae. The treatment of this condition is challenging, and many hypotheses have been proposed. Non-invasive vagus nerve stimulation using slow-paced breathing (SPB) could stimulate both central nervous system areas and parasympathetic autonomic pathways, leading to neuromodulation and a reduction in inflammation. The aim of the present study was to evaluate physical, cognitive, emotional symptoms, executive functions and autonomic cardiac modulation after one month of at-home slow breathing intervention. Methods: 6655 healthcare workers (HCWs) were contacted via a company email in November 2022, of which N = 58 HCWs were enrolled as long COVID (cases) and N = 53 HCWs as controls. A baseline comparison of the two groups was performed. Subsequently each case was instructed on how to perform a resonant SPB using visual heart rate variability (HRV) biofeedback. They were then given a mobile video tutorial breathing protocol and asked to perform it three times a day (morning, early afternoon and before sleep). N = 33 cases completed the FU. At T0 and T1, each subject underwent COVID-related, psychosomatic and dysfunctional breathing questionnaires coupled with heart rate variability and manual dexterity assessments. Results: After one month of home intervention, an overall improvement in long-COVID symptoms was observed: confusion/cognitive impairment, chest pain, asthenia, headache and dizziness decreased significantly, while only a small increase in manual dexterity was found, and no relevant changes in cardiac parasympathetic modulation were observed.

## 1. Introduction

Even though the pandemic era is over, many COVID-19 survivors are still experiencing long-lasting effects from an acute infection, with multifaceted symptoms that vary greatly from person to person. A consensus definition of this condition is not fully acknowledged, but the WHO at the end of 2021 defined “long COVID” as the persistence of one or more symptoms over 12+ weeks from the acute infection, which cannot be explained by an alternative diagnosis [1].

The prevalence of this phenomenon is difficult to evaluate due to the differences among studies in case definition, time elapsed since COVID-19, symptoms reported, or confirmed diagnoses, to cite some issues. Morever, a metanalysis of 41 studies estimates the European prevalence of this phenomenon to be around 44% of previously infected subjects [2]. A wide variety of symptoms have been recognised, among which neurological, cognitive and psychiatric ones are commonly the most reported [2,3,4]. The presence of these symptoms in the majority of these cases does not preclude the possibility of returning to work. However, they do render the performance of one’s duties more challenging, particularly in activities that demand both physical and emotional commitment and the caring for others, as is the context of healthcare workers (HCWs) [5,6]. Improving the health condition of these individuals is crucial for multiple reasons, impacting both individual well-being and public health, and to this end, many not mutually exclusive treatments options have been claimed [7,8].

In this regard, while numerous RCTs are aimed at investigating the pharmacological efficacy of various classes of drugs in reducing symptoms (anti-inflammatories, antidepressants, antihistamines, anticlotting, and steroid medication, among others) [9], there is also a growing number of complementary treatments that are currently being evaluating [10]. Such options include physical techniques, supplementation, regenerative treatments and electrical stimulation—to cite some—that can be beneficial as the main options when a patient is unresponsive or has not responded adequately to pharmacological interventions or can be used as concurrent add-on therapies in other circumstances, possibly leading to an enhanced efficacy of the medical treatments.

In long-COVID subjects, autonomic imbalance has been identified as a potential underlying mechanism [11,12,13]. In the majority of cases, this manifests as a parasympathetic impairment [14,15] and disrupted vagal signalling, particularly following a prolonged course of the acute infection. This is also confirmed by the findings of brain functional MRI studies on adult and paediatric long-COVID subjects, which revealed a decreased metabolism in the central autonomic network (CAN) [16] mainly composed of the limbic system, and also in the brain stem, where the vagus nerve nuclei originate [17,18]. Similarly, deficits in executive functions have been observed in these subjects, as the nature of these processes necessitates a robust exchange of information within the neural network between the cortex and subcortical regions, as well as between the two hemispheres of the brain [19]. Moreover, the persistence of a systemic inflammation could be a consequence of an effectiveness decline in the cholinergic anti-inflammatory pathway (CAP), led by the vagus nerve. In this case, the release of Ach quenches the excess of cytokine release from the mastocytes, which has detrimental effects on the body’s recovery after an acute infection [20,21]. For all the aforementioned reasons, vagus nerve stimulation (VNS) has been proposed as a potential treatment option in long-COVID subjects [22,23].

Up to date, few studies have evaluated this approach, mainly though the electrical transcutaneous stimulation of its auricolar branch [24,25], with encouraging results on fatigue and cognitive symptom reduction in a restricted number of individuals. These approaches are minimally invasive yet necessitate the acquisition of a specific device, training duly the patient for its correct utilisation, and has the limitation that it can only stimulate the VN unilaterally.

An alternative, entirely non-invasive method of stimulating this nerve is through breathing modulation (respiratory VN stimulation—rVNS), which is costless and can be performed by the patient at any time and in any location, on the condition that they have received adequate training in its correct and effective execution.

Respiratory VNS is achieved through the slowing down of breathing frequency to six cycles/min (corresponding to the resonant frequency (~0.1 Hz)), which rationally relies on the neurovisceral integration model.

According to this acknowledged theory [26,27], the CNS and autonomic system are bidirectionally connected and can interact with each other both via equally efficient top–down or bottom–up stimuli [28]. In this case, deep breathing at a low respiration rate is the bottom–up trigger and, compared to tVNS, permits a bilateral stimulation of the VN afferent and efferent fibres [29].

When a subject breathes at six cycles/min, projections of central rhythm generator neurons stimulate the nucleus ambiguus (NA), from which a cascading loop activation is established, which is ultimately responsible for the beneficial effects described in the literature. Vagus cardiomotor efferent fibres originate from NA and reach the heart, thereby inducing a bradycardic effect through an enhanced release of Ach. The aortic bulb and carotid wall baroreceptors record an increased cardiac output, which happens with a 5 s delay from the Ach release. Through this delay, the system achieves temporal coherence between respiratory phases, cardiac phases and blood pressure. Moreover, the vagal afferent fibres departing from the baroreceptors stimulate in turn the medullar nucleus of the solitary tract. From this point, the vagal afferent fibres send widespread signals directly to the cingulate and insular cortex and indirectly to the subcortical limbic system in a longer and more powerful way compared to when spontaneous breathing occurs. This activation of interoceptive areas is responsible for the cognitive, emotional, executive and behavioural changes that can be observed [30,31].

The aim of the present study was to evaluate, in a group of healthcare workers affected by long-COVID symptoms, the effects of one-month slow-paced breathing intervention on perceived cognitive functions, cardiac autonomic modulation, and executive functions.

## 2. Materials and Methods

### 2.1. Study Design and Ethical Aspects

This interventional study was approved by the FVG’s Unite Research Ethics Committee (approval number 245_2023H, ID 17328). All participants signed an informed consent form before being included in the study; the study was conducted in accordance with the principles of the Declaration of Helsinki.

### 2.2. Participants Enrolment

In early November 2022, all ASUGI employees (n = 6655) were informed via a company email about the nature and purpose of the study. Enrolment was then carried out for each worker who expressed interest in participating in the project and who met the inclusion criteria. Inclusion criteria for the long-COVID HCW group were as follows: previous COVID-19 confirmed by RT-PCR on a nasopharyngeal swab (NPS) and the persistence or new onset of symptoms (1 or more of those listed by [32] beyond 12 weeks after NPS negativisation and still present at baseline). Exclusion criteria were as follows: previous COVID-19 and symptoms still present but less than 12 weeks. Inclusion criteria for the control group of HCWs were as follows: having been regularly tested for SARS-Cov-2 by the NPS in accordance with the hospital safety protocol for the pandemic period and having always been found negative.

### 2.3. Data Collection

Each participant underwent a series of questionnaires and instrumental assessments, which are described in detail below, in order to assess the following: demographical data, long-COVID symptoms, respiratory dysfunction, manual dexterity and cardiac autonomic modulation.

General health and long-COVID symptom questionnaires: Data on age, sex, weight and height, smoking habits, SARS-Cov-2 vaccination status, comorbidities, pharmacotherapy and job descriptors were collected. The symptoms of long-COVID were assessed according to the ones described by the Istituto Superiore di Sanità. [32].Psychosomatic symptoms using the M.U.S. and Distress questionnaires related to autonomic impairment: This tool allows the investigation of “medically unexplained symptoms”. It consists of 39 items exploring the physical, psychological, cognitive and emotional domains. Having more than 6 of these symptoms has been associated with low heart rate variability (RMSSD and SDNN values) and an impaired autonomic nervous system [33,34]. Each item can be rated by the patient on a scale from 0 (no symptom) to 10 (highest intensity). The final score is calculated by the sum of the intensity of each item.Dysfunctional breathing symptoms through Nijmegen questionnaire (NQ): Dyspnoea has been described as a common symptom in long-COVID subjects, but it may derive from multiple causes, among which psychological ones may also play a role. This tool is composed of 16 items, each of which can be scored from 0 to 4 by the participant. It has been used to assess dysfunctional breathing patterns [35].Manual dexterity assessment through Purdue Pegboard test manual dexterity (PPT): After instructions, a short demonstration was delivered to the subject. Subsequently, each participant had the possibility to have a trial before the administration of the evaluated test. The procedure comprises a series of steps that must be carried out in a specific order. The subject must pick up pins placed in a bowl on the corresponding side of the tested hand and try to place as many pins as possible in the prepared holes within 30 s. The task is carried out first with the dominant hand, then with the non-dominant hand, and finally, with both hands at once (three separate tasks). The assembly test is then performed, which is composed of a standardised sequence of assembly of pegs, washers and collars with both hands in one minute. The total score is then calculated adding the 1 to 4 task scores together [36,37].Cardiovascular autonomic function assessment through photoplethysmography (PPG Stress Flow^®^, BioTekna Co, Italy): This device allows the detection of the digital pulse waves produced by the systolic contractile force of the heart. After 20 min of acclimatisation to the experimental environment (room temperature 22–24 °C; relative humidity 40–60%) while the subjects are seated in a comfortable chair with their forearms resting on the table, the procedure starts. An infrared probe is placed on the II finger of both hands separately, and the acquired optical signal is then processed by means of appropriate algorithms in order to derive the ECG signal. The detailed procedure has been described elsewhere [38]. The equipment used for the study has been authorised by the Italian Ministry of Health (CND Z12040113). Patients underwent the study protocol in the morning. Measurements were taken for 5 min during spontaneous breathing and then for 5 min during slow breathing (6 breaths/min, i.e., 0.1 Hz).Slow-paced breathing protocol: Each long-COVID participant was taught how to perform resonant slow-paced breathing (SPB) during the initial inpatient assessment, a manoeuvre which elicits a cardiovagal modulation. The breathing cycle was the same for all participants and was composed of 2 s. inspiration, 3 s. holding, 2 s. expiration, and 3 s pause (6 breath per minute, i.e., 0.1 Hz), repeated over a 5 min period. This breath frequency is acknowledged to be capable of maximising the temporal coherence between blood pressure, respiratory and cardiac phases, leading to higher cardiac oscillations, which in turn deliver the central and peripherical benefits [30]. The cases were then instructed to perform it three times a day (morning, early afternoon and before bedtime) for one month using a mobile video tutorial. The researchers contacted them once a week to check that they were following the instructions. Figure 1 shows the flowchart of the study.

### 2.4. Statystical Analysis

The Shapiro–Wilk test was employed to assess the normality of the distributions. Normally distributed continuous data were reported as means (SD) and compared by Student’s t-test or the paired t-test if independent or correlated, respectively. Medians (IQR) were used to report not-normally distributed data, which were contrasted by the Wilcoxon Rank sum test or the Wilcoxon Sign Rank test if independent or correlated, respectively. Numbers (percentages) were used to report normally distributed ordinal data, which were compared using the χ^2^ test or the McNemar chi-square test if independent or correlated, respectively. Not-normally distributed ordinal data were compared with Fisher’s exact test or the McNemar exact test if independent or correlated, respectively. All analyses were conducted using Stata^®^ software version 16 (StataCorp LP, College Station, TX, USA). All *p*-values are two-tailed. Values < 0.05 were considered statistically significant. 

## 3. Results

N = 58 long-COVID subjects were enrolled in the study, out of which N = 33 completed the study and their data were included for the final analysis. N = 53 controls age and sex matched were used as the baseline comparison for MUS and dysfunctional breathing symptoms, HRV and manual dexterity parameters; the results are described elsewhere [37,38]. Moreover, 56% of the cases complained of symptoms for 12+ months, among which cognitive impairment (memory and concentration problems, confusion), a feeling of tightness/anxiety/chest pain, asthenia, dyspnoea and sleep disorders ranked highest (an overall assessment based on questionnaires). None of them had ever been hospitalised for COVID-19 and did not differ significantly from healthy controls in terms of comorbidities (before and after COVID-19) and medication use (for the full results, see [15]). Regarding autonomic symptoms, assessed by M.U.S. and Distress questionnaires (Appendix A), cases showed a significantly higher total number (7.5 (IQR 5-11) vs. 3 (IQR 1-5)) and intensity of symptoms compared to the controls (in 26 out of 38 items (68%)) among physical, cognitive and emotional ones.

After one month of home intervention, N = 33 HCWs completed the FU and received a second inpatient assessment. In total, 63% of them reported that they adhered completely/closely to the protocol, while 36% seldom/never adhered.

The FU population had a median age of 51.5 years (IQR 42-57) and 78% were women. Moreover, one-third were smokers and the median time since the primary infection was 419.5 (IQR 268-730) days.

The prevalence of dysfunctional breathing symptoms was found to have decreased significantly after a one-month intervention, as illustrated in Figure 2. This finding was observed alongside a reduction in the occurrence of other long-COVID symptoms and a decline in the total number of disorders affecting various systems and apparatus (Figure 3a,b and Appendix A).

No relevant changes in the autonomic cardiac modulation were found between T0 and T1, nor in the intensity of psychosomatic symptoms (see Appendix A). The Purdue Pegboard test, assessing manual dexterity, showed only a slight increase in the two-handed assembly task score, approaching statistical significance (*p* = 0.07, Appendix A). This can reflect a marginal enhancement in visuo-motor, coordination and executive skills.

## 4. Discussion

Our study evaluated a group of N = 58 healthcare workers affected by long COVID, of whom N = 33 completed one month of the at-home slow breathing programme and received a second out-patient assessment. Each participant has been evaluated before and after the intervention in terms of physical, cognitive and emotional symptom magnitude, cardiac autonomic modulation during vagal stimulation (slow-paced breathing stimulus), and performances in fine manipulative skills.

Our follow-up population consisted of 78% females, with a mean age of 51.8 years (SD 8.8), who had not previously been hospitalised due to COVID-19, evaluated at a relatively long time since an acute infection of 419.5 days (IQR 269-730). Only a small percentage of cases had restrictions on their fit-for-work judgment (such as night shifts or heavy lifting) and most returned to work in the same position as before COVID-19. They were all vaccinated with one or more doses of COVID-19 vaccine (required by law for health workers in Italy at the time prior to the study) and showed a very high antibody titre compared to never-infected but vaccinated controls. There was no excess of newly diagnosed pathologies in the cases, but in 68% of the items investigating psychosomatic disorders (M.U.S. questionnaire), the cases had a significantly higher intensity of impairment than the controls (physical, cognitive, and emotional areas, but not the behavioural one). In addition, 75% of them had a total number of disturbances ≥6, a datapoint that has been shown to be associated with low HRV values and RMSSD in particular, an indirect parameter of the parasympathetic autonomic component state of health [33,34].

After one month of the breathing intervention, the results showed a significant reduction in symptoms related to cognitive deterioration, headache, asthenia, chest tightness and dysfunctional breathing,

These findings are consistent with the ones of other studies, which tried to stimulate the VN in different ways.

Badran et al. [24]. and Zheng et al. [27] used electrical transauricular VN stimulation (tVNS), demonstrating a reduction in fatigue and cognitive symptoms in a small group of cases, without any further instrumental evaluation.

Corrado et al. [39] and Polizzi et al. [40] used the respiratory stimulation of the VN (rVNS) like we did, finding a significant reduction in perceived fatigue, autonomic symptoms, breathlessness and an enhancement in the ability to focus, stress control, and quality of life and sleep after the intervention, even if they used a different paced breathing shape. The first study was carried out on a small number of patients (*n* = 13) at a mean time of 14.7 (SD 5.8) months after acute infection. An assessment was conducted using a Polar watch coupled with a chest strap and a phone application to record HRV variations during SPB and a battery of questionnaires was used to evaluate autonomic imbalance, disability and quality of life (C19-YRSm, COMPASS-31, WHODAS and EQ5D-5L). The second study retrospectively evaluated the subjective effects (self-reported symptoms and well-being on a Likert scale 1 to 5) of a resonant breathing programme (Meo Health) on *n* = 99 long-COVID anonymous subjects, but no data on the time elapsed since the acute infection were available, nor on drug consumption.

In both cases, the protocol included a 4″: 6″ (inspiration-expiration) breathing pattern, performed twice a day for at least 10 min (up to 30 s in Polizzi) for a total of 1 month, which differed from the breath shape we used (2″3″2″3″, insp., holding, exp., holding, three times a day for one month), but both of them guaranteed a final breathing frequency of six cycles/min.

This particular breathing rate has been used in many other clinical conditions [41] since it is capable of inducing a resonant frequency (~0.1 Hz), which increases cardiac oscillations through the maximisation of the temporal coherence of respiratory, blood pressure, and cardiac phases. Higher cardiac oscillations are then responsible for the peripheral effects, while a longer and stronger stimulation of the limbic system via the afferent fibres of the VN are thought to be at the basis of the cognitive and emotional effects [30].

As regards cardiac autonomic modulation we did not find any relevant change in the intensity of the cardiovagal modulation during slow breathing manoeuvres after the intervention (RMSSD: 33 msn (IQR 23–55) vs. 33 msn (IQR 27–53), pre- and post intervention, respectively, *p* = 0.84 and LF power: 7.4 (IQR 7–8.3) vs. 7.7 (IQR 7–8.2), pre- and post intervention, respectively, *p* = 0.91, with the parameters chosen according to [42]), meaning that a cumulative effect and/or an enhancement in the strength of the cardiac parasympathetic stimulation did not happen in our group. This is in contrast with the results of the only other study, to the best of our knowledge, which assessed this point [39], and that found a significant increase in RMSSD at rest after the intervention (RMSSD: 34.2 msn (SD 19.6) vs. 40.9 msn (SD 29.4), pre- and post intervention, respectively, *p* = 0.048).

As regards executive function, which very often worsens in subjects with long COVID [19,43], we found an impairment in fine manipulative skills at baseline [37], which only marginally improved after the intervention. The test used is based on the speed with which the subject can insert a few pins and carry out a simple assembly in a given time. The activities are performed with both hands separately and then with both hands together. This involves understanding, processing and transferring information between both sides of our brain, visual and motor coordination, and lastly, peripheral muscle activation. These deep brain areas probably only partially have been stimulated by the rVNS and need more training with other tools aimed at improving the underlying networking [44]. No other studies, up to date, are available as regards a pre–post evaluation of manipulative skills in long COVID after a VNS intervention.

Given the consistency of the studies in reducing neurological, cognitive, stress-related and respiratory symptoms and the inconclusive results on peripheral autonomic modulation, it can be hypothesised that the efficacy and/or time required for slow controlled breathing to stimulate the VN central afferent fibres and the relative interoceptive brain areas is less than that required to observe effects on the efferent fibres in modulating vagal cardiac function. This finding could be explained by the fact that VN fibres are mixed and unevenly distributed, being made up of 20% efferent fibres and 80% afferent fibres [30,45].

One of the strengths of our study is the relatively bigger number of patients evaluated after a respiratory VN stimulation compared to other studies which have focused on this topic and that subjects received a multidimensional evaluation. Concurrently, this study was initiated as exploratory; thus, its sample size was calculated based on analogous works on this topic, which were scarce at the time. Moreover, we lack a long-COVID control group. This was due to ethical reasons since all 58 long-COVID subjects at the first evaluation were offered access to the breathing protocol, but the 25 dropouts refused to participate in the program and be re-evaluated after a period of time due to personal, time-related or organisational reasons. In particular, shift workers stated that they could not deliberately stop during a shift to perform controlled breathing if it was scheduled at that time of the day. Regardless, we believe that given that 56% of subjects complained of disorders from 12+ months, it is unlikely that the symptoms would have diminished over the course of those month, even without treatment.

## 5. Conclusions

The results of this study, which evaluated the effects of an at-home vagal stimulation through a six-cycle/min breathing protocol (rVNS), indicated that this approach may be beneficial in reducing the symptoms associated with long COVID. These findings contribute to the expanding body of evidence suggesting that this complementary treatment may be beneficial in alleviating the core symptoms experienced by these individuals, including asthenia, cognitive decline, headaches, and confusion. However, further validation is required to substantiate these findings, ideally through a larger sample size and a randomised controlled trial (RCT) study design with a more extended follow-up period to ascertain the long-term efficacy of the rVNS approach.

## Figures and Tables

**Figure 1 biomedicines-12-02254-f001:**
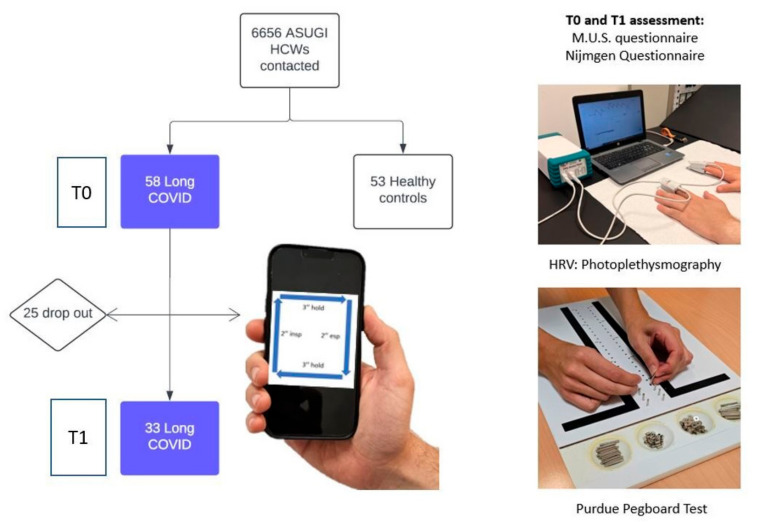
Flow chart of the study, assessment performed and breathing protocol. ASUGI: Azienda Sanitaria Universitaria Giuliano-Isontina.

**Figure 2 biomedicines-12-02254-f002:**
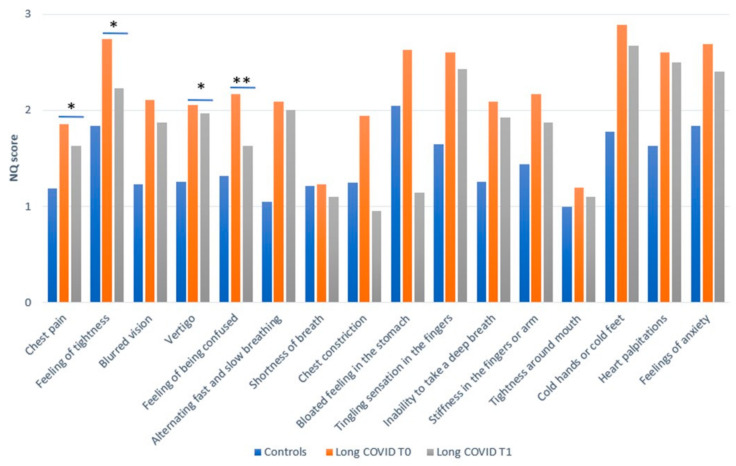
Bar graph of 16-item Nijmegen questionnaire score for control at T0 and long COVID (T0 and T1) who completed the FU. * *p* < 0.05, ** *p* ≤ 0.01, for complete results, see Appendix A.

**Figure 3 biomedicines-12-02254-f003:**
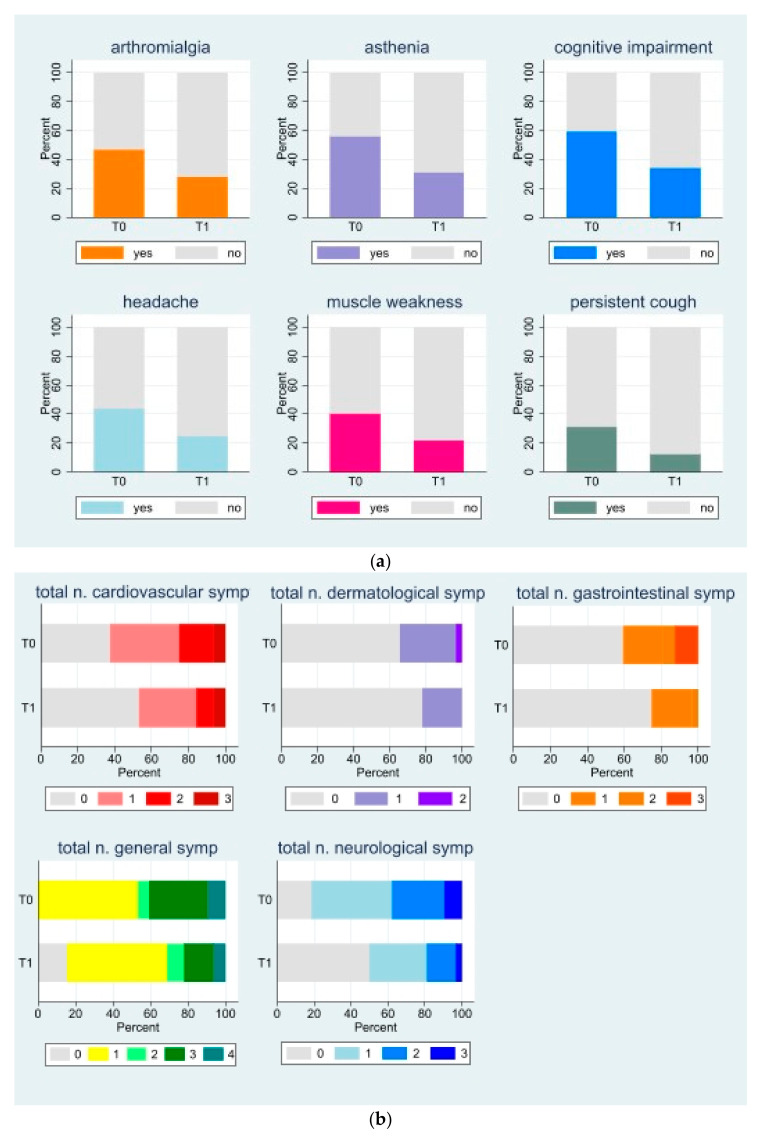
(**a**). Stacked bar graph of long-COVID symptoms, which significantly decreased from T0 to T1. For complete results, see Appendix A. (**b**). Stacked bar graph of total number of long-COVID symptoms by system/apparatus at T0 and T1. See Appendix A for full results.

## Data Availability

The data presented in this study are available on request from the corresponding author. The data are not publicly available due to privacy restrictions.

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
