# Peer review of "Slow-Paced Breathing Intervention in Healthcare Workers Affected by Long COVID: Effects on Systemic and Dysfunctional Breathing Symptoms, Manual Dexterity and HRV"

_biomedicines, 2024, doi:10.3390/biomedicines12102254_

Round 1
Reviewer 1 Report
Comments and Suggestions for Authors
The authors presented interesting results of the study of objective effects of a repeated stimulation of the vagus nerve through a one month at-home slow paced breathing (SPB) intervention in a population of long-Covid Health Care Workers (HCWs). I have some comments:
1) Respect the resonant Slow Paced Breathing (SPB) frequency in the methods. Was the breathing frequency the same for all or was it somehow chosen individually for each subject? Describe the methodology in more detail.
2) In the Introduction it is also necessary to justify in more detail the choice of low-frequency breathing as a method of treatment. The physiological basis of the effects would be well disclosed in the text.
3) I recommend that the Conclusion be given a separate section.
4) 6 breaths/min is a frequency that is often discussed in the context of cardiorespiratory interaction. This is a possible physiological mechanism of action that the authors are exploring. It's not a bad thing to discuss. Various phenomena contribute to the features of cardiorespiratory interactions: resonance responses, phase and frequency locking, synchronization and coordination, etc.
5) Were features of the course of Covid-19 (severity, etc.) taken into account when analyzing autonomic control after Covid-19? This aspect is worth more detailed discussion, also in the form of a possible limitation of the study.
Author Response
|
Thank you very much for taking the time to review this manuscript. Please find the detailed responses below and the corresponding revisions/corrections highlighted/in track changes in the re-submitted files. |
||
|
|
||
|
2. Questions for General Evaluation |
Reviewer’s Evaluation |
Response and Revisions |
|
Does the introduction provide sufficient background and include all relevant references? |
R1: Must be improved
|
Thanks for the suggestion, the introduction has now been expanded to include a more in-depth overview of the slow breathing technique and its potential effects (lines 99-120). Relevant references have been added |
|
Is the research design appropriate? |
R1: Yes
|
// |
|
Are the methods adequately described? |
R1: Must be improved
|
The method section has been improved in each subsection, with more details especially as regard the breath shape and frequency used (lines 224-232) and also as regard the Dysfunctional breathing questionnaire used (lines 183-187). |
|
Are the results clearly presented? |
R1: Can be improved
|
Results have been improved with a better characterization of the studied population (lines 263-269, and 291-295) |
|
Are the conclusions supported by the results? |
R1: Can be improved
|
A separate section for the Conclusion has been added, contextualizing the findings of the study (lines 470-479) |
|
3. Point-by-point response to Comments and Suggestions for Authors
Reviewer 1:
Comment 1: Respect the resonant Slow Paced Breathing (SPB) frequency in the methods. Was the breathing frequency the same for all or was it somehow chosen individually for each subject? Describe the methodology in more detail.
|
||
|
Response 1: Thank you for pointing this out, it was not so clearly explained in the method section. Now a description has been added from line 225 to line 232 to better define the breathing shape and cycle used, which was standard for each participant.
Comment 2: In the Introduction it is also necessary to justify in more detail the choice of low-frequency breathing as a method of treatment. The physiological basis of the effects would be well disclosed in the text. Response 2: Thanks for the suggestion, now the introduction has been fully revised (lines 49-124) and the rationale of the slow paced breathing used to stimulate the vagus nerve is better explained (lines 99-120)
Comment 3: I recommend that the Conclusion be given a separate section. Response 3: Thanks for the suggestion. A separate section has been created and the findings of the study has been better contextualized.
Comment 4: 6 breaths/min is a frequency that is often discussed in the context of cardiorespiratory interaction. This is a possible physiological mechanism of action that the authors are exploring. It's not a bad thing to discuss. Various phenomena contribute to the features of cardiorespiratory interactions: resonance responses, phase and frequency locking, synchronization and coordination, etc. Response 4: Yes, you are perfectly right. We added a paragraph in the introduction section (lines 95-120) and in the discussion section (lines 344-349) to better explained the physiological mechanism of action of the slow paced breathing
Comment 5: Were features of the course of Covid-19 (severity, etc.) taken into account when analyzing autonomic control after Covid-19? This aspect is worth more detailed discussion, also in the form of a possible limitation of the study
Response 5: thanks, we better explained the evaluation we did as regards this point. We added a paragraph in methods section to explain the use of MUS questionnaire to evaluate autonomic symptoms (lines 175-182) and in results section (lines 263-269) to highlight the previous course of COVID-19 severity in cases (none was hospedalized) and the comparison with controls as regard new diagnosed comorbidities and medication use. |
||

Reviewer 2 Report
Comments and Suggestions for Authors
Dear Authors,
here my comments.
ABSTRACT: 1. Add the aim of the study at the end of the introduction. 2. When mentioning concepts like "for the first time," avoid synonyms and instead express them clearly (e.g., HRV).
KEYWORDS: I suggest using MeSH keywords for better accuracy.
INTRODUCTION:
- In the aim of the study, please specify what you mean by subjective and objective effects.
- Add a definition and explanation of the use of slow-paced breathing.
Table: check the citations of the Tables in the text and order them
DISCUSSION: 1. I recommend moving the paragraph starting with "Our study evaluated a group" to the beginning of the discussion section.
2. Today, there is a great deal of attention on high-flow therapy and its influence in various clinical settings. Although high-flow therapy and slow-paced breathing have different primary purposes, they may have a useful correlation in certain clinical contexts. Both high-flow oxygen therapy (HFOT) and slow-paced breathing (slow and rhythmic breathing) affect the respiratory system and can have complementary beneficial effects in some clinical settings: improvement of breathing, reduction of respiratory effort...(DOI: 10.1183/23120541.00075-2024)
Author Response
|
Thank you very much for taking the time to review this manuscript. Please find the detailed responses below and the corresponding revisions/corrections highlighted/in track changes in the re-submitted files |
||
|
|
||
|
2. Questions for General Evaluation |
Reviewer’s Evaluation |
Response and Revisions |
|
Does the introduction provide sufficient background and include all relevant references? |
R2 : Must be improved
|
Thanks for the suggestion, the introduction has now been expanded to include a more in-depth overview of the slow breathing technique and its potential effects (lines 99-120). Relevant references have been added |
|
Is the research design appropriate? |
R2: Yes
|
// |
|
Are the methods adequately described? |
R2: Yes
|
// |
|
Are the results clearly presented? |
R2: Yes
|
// |
|
Are the conclusions supported by the results? |
R2: Yes
|
// |
|
3. Point-by-point response to Comments and Suggestions for Authors
|
||
|
Reviewer 2 Comment 1: Abstract. Add the aim of the study at the end of the introduction. 2. When mentioning concepts like "for the first time," avoid synonyms and instead express them clearly (e.g., HRV). Response 1: Done, thanks for the suggestion (lines 22-24, 27, 35-35)
Comment 2: KEYWORDS: I suggest using MeSH keywords for better accuracy. Response 2: Thanks, we found and added/chanhed the following keywords: MUS: unique ID D000071896, long Covid: unique ID D000094024, autonomic effects: unique ID D001337, breathing exercises: unique ID D001945, clinical trial: unique ID D016430, Executive functions: unique ID D056344
Comment 3: INTRODUCTION: 1.In the aim of the study, please specify what you mean by subjective and objective effects. 2.Add a definition and explanation of the use of slow-paced breathing. Response 3: 1. The sentence has been rephrased to gain more clarity (lines 22-23, 121-124). 2. We added a description and explanation of the working mechanismof slow paced breathing in the introduction section accordingly (lines 99-120)
Comment 4: Table: check the citations of the Tables in the text and order them. Response 4: Thank you, the tables are now ordered according to their citation in the text.
Comment 5: DISCUSSION: I recommend moving the paragraph starting with "Our study evaluated a group" to the beginning of the discussion section. Response 5: done, thanks for the suggestion, now the findings are more clear and consequently the comparison with the results of other similar studies.
Comment 6: DISCUSSION Today, there is a great deal of attention on high-flow therapy and its influence in various clinical settings. Although high-flow therapy and slow-paced breathing have different primary purposes, they may have a useful correlation in certain clinical contexts. Both high-flow oxygen therapy (HFOT) and slow-paced breathing (slow and rhythmic breathing) affect the respiratory system and can have complementary beneficial effects in some clinical settings: improvement of breathing, reduction of respiratory effort...(DOI: 10.1183/23120541.00075-2024) Response 6: HFOT could be another rational approach to reduce symptoms in Long Covid subjects, but we didn’t find any preliminary study up to date and we believe it is too far away from the topic discussed in this study. We apologize. |
||

Reviewer 3 Report
Comments and Suggestions for Authors
The authors have proposed the following manuscript: "Slow paced breathing intervention in healthcare workers affected by long COVID: effects on systemic and dysfunctional breathing symptoms, manual dexterity and HRV".
The manuscript is rather succinctly presented, all chapters not presenting enough information about the study in question. It should be mentioned from the outset that the number of subjects chosen (58) of which 33 remain at the end is insufficient to validate the study.
Author Response
|
Thank you very much for taking the time to review this manuscript. Please find the detailed responses below and the corresponding revisions/corrections highlighted/in track changes in the re-submitted files. |
||
|
|
||
|
2. Questions for General Evaluation |
Reviewer’s Evaluation |
Response and Revisions |
|
Does the introduction provide sufficient background and include all relevant references? |
R3: Must be improved
|
Thanks for the suggestion, the introduction has now been expanded to include a more in-depth overview of the slow breathing technique and its potential effects (lines 99-120). Relevant references have been added |
|
Is the research design appropriate? |
R3: must to be improved |
Unfortunately it cannot be changed at this point |
|
Are the methods adequately described? |
R3: must to be improved |
The method section has been improved in each subsection, with more details especially as regard the breath shape and frequency used (lines 224-232) and also as regard the Dysfunctional breathing questionnaire used (lines 183-187). |
|
Are the results clearly presented? |
R3: must to be improved |
Results have been improved with a better characterization of the studied population (lines 263-269, and 291-295) |
|
Are the conclusions supported by the results? |
R3: must to be improved |
A separate section for the Conclusion has been added, contextualizing the findings of the study (lines 470-479) |
|
3. Point-by-point response to Comments and Suggestions for Authors
|
||
|
Reviewer 3: Comment1: The manuscript is rather succinctly presented, all chapters not presenting enough information about the study in question. Response1:Thanks for the comment, that allowed a throughout revision of the manuscript to better valorize the work done. Lines 49-124, 171-232, 263-269, 293-297, 299-392, 471-480)
Comment2: It should be mentioned from the outset that the number of subjects chosen (58) of which 33 remain at the end is insufficient to validate the study. Response2: We thank the reviewer for highlighting this important aspect of our study. We understand the concern regarding the number of patients included in the analysis. Since these studies are exploratory, no specific sample size was planned for this sub-analysis. We encountered several constraints during recruitment, such as limited patient availability, but we ensured that the included patients met all the required criteria to maintain the study's validity. Despite the limited number of patients, we employed rigorous statistical methods to ensure the reliability of our findings. We acknowledge that a larger sample size would provide greater statistical power, which is a limitation of our current work. However, our results are consistent with previous studies on the subject, further supporting our conclusions. We have added several sentences addressing this in the Discussion section on page 10, lines 382-284 and conclusion section lines 477-480. |
||
|
|
||

Round 2
Reviewer 1 Report
Comments and Suggestions for Authors
I approve of the current version of the article.
Reviewer 2 Report
Comments and Suggestions for Authors
Dear Authors,
Reviewer 3 Report
Comments and Suggestions for Authors
The authors have taken into consideration the comments provided and modified the manuscript in such regard therefore I propose the manuscript for publication.